# Biogenic calcium carbonate as evidence for life

Sara Ronca[1], Francesco Mura[2], Marco Brandano[1], Angela Cirigliano[3], Francesca Benedetti[4], Alessandro Grottoli[5], Massimo Reverberi[6], Daniele Federico Maras[7], Rodolfo Negri[4], Ernesto Di Mauro[3] & Teresa Rinaldi[4]

[1]Department of Earth Sciences, Sapienza University of Rome, Italy
[2]CNIS-Center for Nanotechnology Applied to Industry of La Sapienza, 00185, Sapienza University of Rome, Rome, Italy
[3]Istituto di Biologia e Patologia Molecolari, CNR, Rome, Italy
[4] Department of Biology and Biotechnologies, Sapienza University of Rome, Rome,
[5]SARA EnviMob s.r.l., Rome, Italy
[6]Department of Environmental Biology, Sapienza University of Rome, Rome, Italy
[7]Soprintendenza Archeologia, Belle Arti e Paesaggio per la Provincia di Viterbo e l'Etruria Meridionale, Ministero della Cultura, Rome, Italy

*Correspondence to*: Teresa Rinaldi (teresa.rinaldi@uniroma1.it)

**Abstract.** The history of Earth is a story of co-evolution of minerals and microbes: not only numerous rocks arisen from life, but the life itself may have formed from rocks. To understand the strong association between microbes and inorganic substrates, we investigated the moonmilk, a calcium carbonate deposits of microbial origin, occurring in the Iron Age Etruscan Necropolis of Tarquinia, in Italy. These tombs provide a unique environment where the hypogeal walls of the tombs are covered by this speleothem. To study moonmilk formation, we investigated the bacterial community *in* the rock in which the tombs were carved: calcarenite and hybrid sandstone. We present the first evidence that moonmilk precipitation is driven by microbes *within* the rocks and not only *on* the rock surfaces. We also describe how the moonmilk produced within the rocks contributes to rock formation and evolution. The microbial communities of the calcarenite and hybrid sandstone displayed, at the phylum level, the same microbial pattern of the moonmilk sampled from the walls of the hypogeal tombs, suggesting that the moonmilk originates from the metabolism of endolytic bacterial community. The calcite moonmilk is the only known carbonate speleothem on Earth with undoubted biogenic origin, thus representing a robust and credible biosignature of life. Its presence in the inner parts of rocks adds to its characteristics as a biosignature.

## 1 Introduction

Whether other planets witnessed life like what is seen on Earth remains a complete mystery. The search for traces of extra-terrestrial life suffers from the lack of durable and credible biosignatures. Some breakthroughs may happen soon from the current planetary explorations. The NASA Perseverance rover successfully landed on planet Mars in the Jezero Craters in 2021 and has been collecting many specimens. Future missions are planned to retrieve those specimens, although not before 2031, hence opening the possibility to search for evidence of life from the first-ever samples returned from Mars. The European Space Agency (ESA) ExoMars programme is also planning to address the question of whether life has ever existed on Mars.

Earth has evolved through a long process of co-evolution between minerals and microbes (Cosmidis and Benzerara, 2022; Grosch et al., 2015; Cuadros, 2017; Hazen et al., 2008) and terrestrial rocks constitute an ideal system for the investigation of durable signs of life. Carbonates are good candidates as host-rock because, on Earth, they are largely of biogenic origin. Nevertheless, they did not receive the attention needed because of limited and sporadic evidence on Mars: lithologies revealed

the presence of carbonates in the Nili Fossae Region (Ehlmann et al., 2008), at the Mars Phoenix landing site (Boynton, et al., 2009), in the Columbia Hills of Gusev crater (Morris et al., 2010), and in deep rocks exposed by meteor impacts (Michalski et al., 2010). Recently, the analysis of weathering profiles revealed the widespread distribution on Mars of carbonates associated with hydrated minerals, providing evidence of past liquid water presence (Bultel et al., 2019). Carbonates were also detected in asteroids and meteorites, contributing to the understanding of the formation and evolution of our solar system (Lee et al.,

2014; Kaplan et al., 2020; Pilorget et al., 2021; Voosen, 2020).

Carbonate rocks on Earth are of abiogenic or biogenic origin and arose since the early Archean (> 3 giga annum, Ga) Eon when hydrothermal systems were ubiquitous. At that time, the carbonate rocks originated by massive carbonatization, silicification and potassium (K) (±Sodium) metasomatism of intermediate to ultramafic silicate precursors (Veizer et al., 1989). In contrast, Lower Archean marine carbonates are rare, they occur as very thin, discontinuous, and extensively mineralized

beds generally replaced by chert as result of intense microbial iron (Fe) cycling (Pomar, 2020). The 3.4 Ga old stromatolite structures (sensu Riding, 2011) are widely regarded as among the oldest biogenic carbonate production associated with carbon dioxide ($CO_2$) sequestration (Allwood et al. 2006). During the Proterozoic eon (<2.4 Ga), shallow water carbonate production expanded favoring the development of carbonate platforms where abiogenic and biogenic carbonate precipitation took place (Grotzinger et al., 2000). The seawater was supersaturated with both calcite and aragonite as evidenced by the well preserved

pseudomorphs of 'abiogenic' aragonite and calcite (Pomar, 2020). According to Sumner and Grotzinger (1996), the rise in oxygen concentration at 2.2–1.9 Ga led to the removal of $Fe^{2+}$, a strong calcite-precipitation inhibitor, from seawater and resulted in a shift from Archean to Proterozoic carbonates that are dominated by microbial activity (Pomar, 2020). Starting from the Cambrian Period, the benthic microbialites (sensu Riding 2011), the prime carbonate factory since the Late Archean, were still important but progressively decreased, while the biologically controlled carbonates appeared and expanded. The

calcification of sessile, mostly colonial, metazoans and algae promoted the accumulation of biogenic carbonate sediments and the appearance and expansion of reefs (Pomar, 2020).

Carbonates are common constituents of the near-surface Earth crust, although carbonate phases may also occur deep in the mantle. They are compounds formed by the anionic complex, carbonate $(CO_3)^{2-}$ combined with metal ions such as calcium (Ca), magnesium (Mg), iron, manganese, sodium, barium, aluminium, zinc, copper, lead, uranium, or rare-earth elements.

Uncommonly, carbonate phases are hydrated and contain hydroxyl or halogen ions or may include silicate, sulphate or phosphate radicals. Due to the large availability of Ca and Mg in crustal reservoirs (Hartmann et al., 2012), the $CaCO_3$ polymorphs, calcite and aragonite, and dolomite ($CaMg(CO_3)_2$) are the most widespread carbonate minerals, whose formation on Earth near-surface environments is widely related to biogenic or bio-mediated processes (Görgen et al., 2021).

Carbonate biomineralization or organomineralization (sensu Dupraz et al., 2009) results in the formation of several mineral phases, the most common of which are: the $CaCO_3$ anhydrous polymorphs calcite, aragonite and vaterite, the last being a metastable transitional phase, the hydrated forms monohydrocalcite ($CaCO_3·H_2O$) and ikaite ($CaCO_3·6H_2O$), and various amorphous phases (ACC). Moreover, in specific environments (saline lakes, coastal lagoons) (Diloreto et al., 2021; Kaczmarek et al., 2017), microbial activity may promote the formation of dolomite (ordered phase $CaMg(CO_3)$ by passing through the precursor phases of high Mg-calcite (disordered 4–36 mol % $MgCO_3$), disordered dolomite (disordered >36 mol % $MgCO_3$) and proto-dolomite (weakly ordered >36 mol % $MgCO_3$). Carbonate mineral formation seems to proceed from amorphous or disordered phases towards the more stable and ordered forms (Asta et al., 2020); crystal growth and morphologies are controlled by the medium composition, the microbial extracellular polymeric substances (EPS), the Mg/Ca ratio and the presence of other ions. Thus, the mineral phases resulting from abiogenic or biogenic activity are indistinguishable, and the identification of irrefutable biosignature, evidence of past or present life, remains absent (Changela et al., 2021; Javaux, 2019).

Carbonate rocks, being mostly of biogenic origin, could be considered in this perspective as possible biosignature but, even if specific analyses could discriminate between biogenic and abiogenic carbonate rocks (Blanco et al., 2013), the attempt to unequivocally distinguish between carbonates of biogenic or abiogenic origin remains vacuous, especially in a search for evidence for life on other planets. So far, the presence of organic materials in carbonate has not been an incontrovertible indicator of biogenicity (Berg et al., 2014).

Consideration for calcium carbonate minerals as possible biosignatures can be found by the study of a secondary calcite deposit, called moonmilk, formed by nanofibers of calcite, commonly found on karst caves surfaces. Calcium carbonate deposits, consisting of thin fiber crystals, have been observed in various vadose environments (soils, karstic caves and other hypogeal spaces), and an exhaustive classification of calcite fibers morphology is presented in (Cailleau et al., 2009). Although widely studied, the origin of such deposits is still a debated matter, being attributed to physicochemical processes (Kolesàr and Čurlik 2015; Jones & Peng, 2014) or to various biogenic processes (Canaveras et al., 2006; Millière et al., 2019). In the past, inorganic precipitation mechanisms were thought, either to be controlled by climatic variations, as in the case of the *Caverne de L'Ours* (Lacelle et al., 2004) in Canada, or from the heterogeneous nucleation of calcite from supersaturated fluids as in the Staloti cave (Borsato et al., 2000). More recently, many authors discussed the biogenicity of the moonmilk (Baskar et al., 2011; Braissant et al., 2012; Cacchio et al., 2014; Kondratyeva et al., 2020; Maciejewska et al., 2017; Portillo & Gonzalez 2011) reporting that the moonmilk precipitation is promoted by the metabolic activities of a microbial community living in environments rich in calcium content (Banks et al., 2010; Cailleau et al., 2009; Cirigliano et al., 2018; Portillo et al., 2011). Nevertheless, *direct* evidence that bacteria promote the precipitation of nanofibers of calcite is still lacking. This evidence is essential to define the moonmilk as a *bona fide* biosignature.

Recently, the moonmilk speleothem was discovered in the hypogeal ancient Etruscan tombs of the Monterozzi Necropolis (Tarquinia, Central Italy) (Cirigliano et al., 2018; Tomassetti et al., 2017). This finding provides a unique opportunity to compare the moonmilk which covers the walls and ceilings collected from 12 tombs excavated in two types of rock, the

calcarenite and hybrid sandstone. We provided insight into the formation of moonmilk, which can occur rapidly, i.e. between 10 and 50 years, and we reported that this speleothem originates from, and harbours, a microbial community able to induce carbonate precipitation (Cirigliano et al., 2021a; Cirigliano et al, 2021b). Here, we propose that the nanofiber calcite deposit (moonmilk) developing inside the rocks (calcarenite and sandstone) is promoted by a microbial community and by the physio-chemical features of the host rock. We present an example of an ongoing symbiotic co-evolution between rocks and microorganisms: the moonmilk contributes to the evolution of the rock rich in calcium carbonate, while the physio-chemical features of the host rock shape the resident microbial community which induces the moonmilk deposition.

## 2 Materials and Methods

### 2.1 Site description and sampling

Samples representative of the bedrock were collected from the ancient Etruscan Necropolis of Tarquinia, a UNESCO World Heritage site (Viterbo, Italy), in which more than 200 painted hypogeal tombs (dated from the 7th to the 2nd century B.C.) were discovered. The tombs were excavated in a sedimentary bedrock belonging to a middle to upper Pliocene formation known as *Macco* consisting of yellowish bioclastic calcarenites interbedded with hybrid sandstones (Supplementary Fig. 1 shows the map of the Necropolis in Tarquinia and the sampling locations). From each location, rock samples were collected and kept in plastic bags on ice and transported to the laboratory for analyses. With sterile hammer and chisel, surface material, samples from outdoor and indoor rocks were first removed to a depth of 3 to 5 cm. The interior of the rock samples was processed for DNA extraction or geological experiments.

### 2.2 Media and growth condition for calcium carbonate organomineralization

To study the organomineralization in laboratory, bacterial strains cultures are maintained in LB (Luria Bertani) liquid medium (1% Bacto-triptone, 0.5% di yeast extract, 0.5% NaCl and 0.1% NaOH 1N). The carbonatogenic activity of bacterial strains has been assessed (Supplementary Fig. 9) in solid complete medium YPDuc plates, containing 1% Bacto-peptone, 1% yeast extract, 2% glucose, 4% urea, 2.5% $CaCl_2$ and 2% agar, adjusted to pH 8.0. To assess the carbonatogenic activity of the microbial community (Supplementary Fig. 10), 0,1 gr of crushed calcarenite was inoculated in BPuc medium (0.72% Bacto-peptone, 4% urea and 2.5% $CaCl_2$, pH8.0) for one week at 28°C, at 160 rpm/min. The experiments were performed in three technical and three biological replicas. These media induce the metabolism of urea of the microorganisms, resulting in a fast organomineralization in plates or in liquid medium.

## 2.3 DNA extraction procedures and sequencing of rock and moonmilk samples

For each sample, rock material was collected aseptically using a sterile rock hammer or chisel and stored in sterile collection bags. All samples intended for DNA extractions were collected by discarding the top 3-5 cm layer and then crushing the inner part with a sterile rock hammer and further reducing it to a powder by grinding with a sterile mortar and pestle. Genomic DNA extraction was performed using the DNeasy PowerMax Soil Kit (QIAGEN) following the manufacture manual method using about 10g of collected material. Spectrophotometric quantification was performed using a ThermoScientific NanoDrop spectrophotometer (Thermoscientific), and DNA purity was assessed through the evaluation of 260/280 and 260/230 absorbance ratios. PCR amplification was performed on about 50 ng of DNA of each sample as described by Grottoli et al., (2020). Amplicon sequencing was performed on PCR products deriving from the 16S rRNA samples regions. PCR products were sequenced through Oxford Nanopore Technologies (ONT) by a MinION sequencer. A total amount of 200 ng of PCR products for samples were used for sequencing. Rapid barcoding of PCRs was carried out following the protocol released by ONT (SQK-RBK004) and sequenced by a Flongle flocell (FLO-FLG001). Total run produced ~15 Mb of data (~34 k reads) including: ~2.8 Mb (~4 k reads) for Moonmilk of *Tomba Maggi 2*, ~2.8 Mb (~4 k reads) for Moonmilk of *Tomba degli Scudi*, ~2.7 Mb (~11 k reads) for hybrid sandstone and ~6.4 Mb (~15 k reads) for calcarenite, respectively. Kraken2 system (Wood et al., 2019) produced a taxonomic classification of 98.83%, 96.65%, 44.02% and 62.29% of total reads for each sample, Moonmilk of *Tomba Maggi 2*, Moonmilk of *Tomba degli Scudi*, hybrid sandstone and calcarenite, respectively.

## 2.4 Petrophysical analysis

Polarized light microscopy (PLM) observations of rock samples and moonmilk speleothems have been carried out on thin-sections at the Dipartimento di Scienze della Terra of Sapienza University of Rome by means of Olympus BX50 (Japan) and Zeiss Axiophot (Germany) transmitted polarized light microscopes, under plane-polarized light and crossed polars, at 25×, 40×, 100 ×, 200 ×, 400 × magnifications. Microscopic images were obtained by using a digital camera. Polished thin sections (28 mm x 48mm) were prepared from the samples vacuum impregnated with epoxy resin before cutting and thinning. Density and porosity were measured using Ultra-pyc 5000 helium pycnometer from Anton Paar with an accuracy of 0.02% and repeatability of 0.01%. Bulk density was obtained by dividing the dry mass of sample by its total volume. Grain density results from the calculation of the mass / measured volume ratio of the pulverised matrix. Both total and effective (open) porosity was measured. The latter was obtained by dividing the difference between the geometric volume and the volume measured by pycnometer of the sample (Ruggieri and Trippetta, 2020; Trippetta et al., 2020). All laboratory measurements were made in the Earthquake Physics Laboratory at Sapienza Earth Sciences Department. Bulk rock major and trace element compositions were obtained by lithium metaborate/tetraborate fusion ICP-AES and ICP–MS, at Activation Laboratories Inc. (Ontario, Canada) according to the Code 4Litho code package on solutions prepared with lithium metaborate fusion. Loss on Ignition (LOI) was measured according to standard gravimetric procedures. Details on the precision and accuracy of the analyses are

reported in www.actlabs.com. The calcium carbonate content was assessed by gasometric measurements using a Dietrich–Fruhling calcimeter measuring one gram of the bulk sediment following Siesser et al., (1971).

## 2.5 Scanning Electron microscopy analysis

Scanning Electron Microscopy (SEM) was performed on the Moonmilk samples and on the rock thin sections using a Field Emission Scanning Electron Microscopy (FESEM) Zeiss Auriga 405, with a chamber room that maintains a pressure of about $10^{-5}$ to $10^{-6}$ mbar. Before mounting the samples inside the microscope, the specimens were coated with 20 nm of chromium using a Quorum Q150T sputter. Chromium has a high X-ray Kα value (5.145 keV), so does not interfere with lighter elements during the EDX analysis. EDX spectra were obtained using a Bruker Quantax detector in point mode for 30 s, with the electron microscope acceleration voltage set at 10 kV and working distance of 6 mm to optimize the number of the incoming X-ray signal.

## 3 Results and Discussion

### 3.1 Lithology, mineralogy and geochemical characterization of the bedrock

The area of the Monterozzi Necropolis offers a favourable chance to investigate the role of bedrock in the genesis of the moonmilk. To this purpose, we analysed samples of bedrock taken from different areas of the Necropolis with special focus on the sites where *Tomba dei Vasi dipinti*, *Tomba Maggi 2, Tomba delle Pantere* and *Tomba degli Scudi* were carved (Supplementary Fig. 1). The hypogeal tombs of the Monterozzi Necropolis, located on a flat elevated area, are excavated within a sedimentary substrate of Middle to Upper Pliocene age known as *Macco* formation consisting of two main lithofacies showing lateral and vertical heteropic relationships. *Macco* s.s. lithofacies is a bioclastic calcarenite represented by packstone to rudstone and floatstone composed of small volume (<5-10%) of micrite matrix, coralline algal branches, bryozoans, bivalves (pectinids and oysters), echinoids, benthic foraminifers, and skeletal debris. Rare non-carbonate grains are present. In the inner walls of the intergranular voids microsparite cement precipitation and/or recrystallization often occurs in phreatic marine environment. The second lithofacies is represented by a poorly cemented, crudely stratified, hybrid sandstone. It is fine to medium grain-sized and grain-supported, with small amount of micrite matrix (< 10% vol) and nearly devoid of carbonate cement. This lithofacies is characterised by abundant bioturbation, the skeletal assemblage is dominated by small benthic foraminifers, echinoids, serpulids and bivalves (mainly oysters). Planktonic foraminifers are common. The terrigenous fraction mainly consists of monocrystalline grains of quartz, sedimentary lithoclasts, and subordinate detrital micas and feldspar along with rare glaucony grains and opaques. Calcimetric analyses from the two lithofacies revealed that calcium carbonate contents range from 90 to 98% in the Macco s.s. calcarenite and from 49 to 59 % in the hybrid sandstone. Moreover, whole rock major oxide and trace element compositions highlight the geochemical difference between the two lithofacies (Table 1), mostly related to the higher proportion of terrigenous fraction in the hybrid sandstone. Helium pycnometry revealed a high open porosity in both lithofacies (~43% for Macco s.s.; ~42% for hybrid sandstone). The *Macco* s.s. lithofacies shows a dominance

of vuggy porosity, abundant intraparticle, interparticle and mouldic porosity. The main porosity of hybrid sandstone is represented by interparticle porosity, rarely by mouldic porosity. Thus, despite the differences in composition, the calcarenite and the hybrid sandstone show two essential characteristics required for moonmilk formation: high calcium content, which activates the microbial metabolism leading to organomineralization, and high porosity, necessary for the exchange of fluids and nutrients in an oligotrophic environment, also providing the space for microbial colonization. Noteworthily, the characteristics of bedrock porosity (vuggy and moldic) indicate that the dissolution processes prevail on those of inorganic carbonate precipitation; indeed, meteoric cements as well as speleothems in the largest cavities are absent. Moreover, the bedrock where the tombs are carved is located in the shallow vadose zone (few meters below the surface). The high magnesium calcite, the dominant mineralogy of the main components, the coralline algae, is a metastable mineral phase of calcite. This phase upon exposure to meteoric water dissolves partially increasing the availability of $Ca^{2+}$ for the microbial metabolism and for the biogenic carbonate precipitation. It is to be noted that the area under investigation lies on a high flat relief, where the infiltrated water mostly derives from rainfall, without any groundwater input. Consequently, the infiltrated water can be reasonably assumed undersaturated with respect to calcite.

### 3.2 First report of inner location of moonmilk

Mostly, moonmilk biogenic deposits develop indifferently within the two distinct lithofacies constituting the bedrock of the Etruscan necropolis of Monterozzi (Mura et al., 2021). Fig. 1a shows an example of hypogeal walls of tombs carved in calcarenite and hybrid sandstone, and the walls covered by moonmilk in the *Tomba degli Scudi* and *Tomba Maggi 2* (Fig. 1b). The moonmilk layer originated from hybrid sandstone is thinner than the one observed on a calcarenite substrate, but the scanning electron micrographs of the moonmilk sampled from the walls of *Tomba degli Scudi* and *Tomba Maggi 2* showed the same nanofiber structure (Fig. 1c). X-ray powder diffraction analysis (XRD) revealed that moonmilk is composed of calcite (Mura et al., 2020).

So far, the moonmilk has only been considered as a deposition covering rock surfaces (Borsato et al., 2000), but the analysis with transmitted polarized light microscopy of thin sections showed that the moonmilk is present *inside* the calcarenite in the vuggy and moldic porosity of the calcarenite in *Tomba Maggi* 2 (Fig. 2a) and in intergranular and moldic pores of the hybrid sandstone bedrock collected inside the *Tomba degli Scudi* (Fig. 2b). The presence of moonmilk inside the rocks is a general phenomenon, because it is observed in all samples, irrespective of the type of rock, calcarenite or hybrid sandstone, and collection site (outdoor or indoor) (Supplementary Fig. 2, 3 and 4). These results are also supported by the discovery of moonmilk deep inside a calcarenite rock sampled at the entrance of the *Tomba dei Vasi Dipinti* (Supplementary Fig. 5). The analysis of the rock substrate sampled outdoor the *Tomba dei Vasi Dipinti* also suggests that the moonmilk may be contributing to the authigenic carbonate growth in the host rock, covering the inner walls of the voids (Fig. 3).

### 3.3 Co-evolution of rocks and microorganisms

If the moonmilk observed *inside* the rocks (a location that was not reported before) is of biogenic origin, traces of organomineralization would be expected. Indeed, the SEM analysis on the thin sections of calcarenite sampled outdoor of the *Tomba dei Vasi Dipinti* revealed many structures corresponding to organomineralization (Fig. 4 and Supplementary Fig. 6). Bacterial organomineralization was also detected in the sandstone sampled indoor the *Tomba degli Scudi* (Supplementary Fig. 7) and in calcarenite sampled outdoor of the *Tomba delle Pantere* and indoor of *Tomba Maggi 2* (Supplementary Fig. 8), suggesting that this is a common phenomenon. Such bacterial organomineralization is also known as "*entombment*" and it is easily observed in laboratory settings when bacterial strains are subjected to environmental conditions favoring calcium carbonate precipitation (Supplementary Fig. 9a, b, c). In plates, the precipitation of calcium carbonate occurs even at a considerable distance from the bacterial colony, possibly by the diffusion of extracellular enzymes known to be involved in calcium carbonate metabolism (Dhami et al., 2014; Rodriguez-Navarro et al., 2019) (Supplementary Fig. 9d).

It remains unclear how the moonmilk nanofibers are produced in natural environmental conditions because to date it has been impossible to reproduce its formation in the laboratory. In fact, bacterial strains cultured from rocks represent only a negligible fraction of the total microorganisms present in the rocks. Instead, the entire microbial community, with a metabolism that sustains the growth in the rock environment, is needed to precipitate and/or dissolve calcium carbonate. Indeed, under laboratory conditions, we have evidence that the grinded calcarenite, with its entire microbial community, when present in a medium containing urea and $CaCl_2$, produced calcite. In the same conditions, calcium carbonate is not produced with sterile (autoclaved) grinded calcarenite (Supplementary Fig. 10) (Benedetti et al., in review). These and previous results (Banerjee, & Joshi, 2014), showed that inactivated (dead) cells were unable to precipitate $CaCO_3$ in laboratory, suggesting that cells need to be metabolically active for calcification and that cell structure alone is not sufficient to promote bioprecipitation. The most studied bacterial metabolism for $CaCO_3$ precipitation is the ureolytic metabolic pathway. This process involves the production of carbamate ($NH_2COOH$) by urea hydrolysis, which spontaneously hydrolyses to form ammonia ($NH_3$) and carbonic acid ($H_2CO_3$). These products react with water to form carbonate ($CO_3^{2-}$), ammonium ions ($NH_4^+$) and hydroxyl ions ($OH^-$), finally resulting in an increase of pH. In an alkaline environment, the presence of calcium ions and the bacterial cells, as nucleation site, allows the precipitation of calcium carbonate (Anbu et al., 2016; Hammes et al., 2002). Thus, it is not the presence of the bacterium alone, as a structure, that provides evidence of precipitation, but also high pH, metabolism, and negative membrane charge. In laboratory conditions (Supplementary Fig. 9), if urea and calcium are present in the medium, the mechanism is stimulated/accelerated.

In natural environments, bacteria are centers of nucleation for nanofibers formation. They do not control the mineralization process directly but induce the precipitation of calcium carbonate by changing the chemistry of the environment as a consequence of their metabolic activity, and also serving as nuclei for crystallization. This mechanism is a result of pH increase, a direct effect of the negatively charged bacterial surface, and the presence of a metabolic ureolytic process (Omoregie et al., 2021).

Overall, our results underscore the role of microorganisms in promoting moonmilk deposition, contributing to the rock formation processes. Nevertheless, to propose the moonmilk as part of a geological process, the microbial communities of the rocks and those contributing to the moonmilk deposition should have a similar composition. Aiming to identify the rock microbial communities, samples from calcarenite and hybrid sandstone were analyzed together with the corresponding moonmilk samples from *Tomba Maggi 2* and *Tomba degli Scudi*. The results of the 16S SSU rRNA amplicon sequencing showed a high abundance of Actinobacteria, Bacteroidetes, Cyanobacteria, Firmicutes and Proteobacteria (Fig. 5 and supplementary data). Of note, the Firmicutes phylum is abundant and several members, such as the *Lysinibacillus* genus, have extremely high urease activity and therefore greatly enhance carbonate precipitation (Banerjee and Joshi, 2014; Benedetti et al., 2023; Zhu and Dittrich 2016), see also Supplementary Fig. 9d.

Bacterial community diversity was measured by inverse Simpson index and Shannon index for moonmilk (*Tomba degli Scudi* and *Tomba Maggi 2*) and rocks (calcarenite and hybrid sandstone). The indices do not show any significant differences between the samples (Mann Whitney test, $P > 0.05$) (Supplementary Fig. 11). These results show that in moonmilk and rocks the microbial composition is similar, irrespective of rock type (calcarenite and hybrid sandstone) or the environment where the samples were collected (outdoor or indoor). It should be noted that 16S SSU rRNA analysis does not provide information about metabolic activity, thus these data do not identify microorganisms that are active in $CaCO_3$ deposition, but the overall data demonstrate that the endolytic community of the rocks is promoting moonmilk deposition; the results presented also revealed the presence of organomineralization and calcite nanofibers that originate from bacterial entombment, not only on the surface, but inside the rocks. The presence of a resident microbial community deep within the rocks, possibly evolved with the rocks through geological time. Therefore, no habitat should be considered as *extreme* for the resident microbial community, and the rocks should not be considered as a 'refuge' for escaping extreme environmental conditions. Biological research should focus on microbial community evolution with respect to the geologic substrate in which they are living, considering the natural co-evolution of microbes and rocks, and possibly abstaining from the consideration of the microbial metabolism as an adaptation to adverse environmental conditions.

### 3.4 Moonmilk as a biosignature

The search for traces of extra-terrestrial life is a complex task often ending with inconclusive results. The co-evolution of minerals and microorganisms has implications for the quest of evidence for life on other planets. The discovery of minerals of undisputable biological origin, rather than organic remains, may provide the most robust signs of biological activity (Hazen et al., 2008). Coevolution of life and minerals throughout Earth history lays the foundation for an inclusive search for the presence of life, because not only rocks arose from life but also because life itself may have formed from rocks (Bizzarri et al., 2021; Marshall, 2020; Saladino et al., 2018). Thousands of earth's minerals owe their existence to the development of life on the planet and calcium carbonate phases that are massively produced on earth by microorganisms, are the best example (Hazen et al., 2008).

In this work, we have focused on the calcium carbonate nanofibers (moonmilk). Given the tight association and co-evolution between rocks and microbial communities that results in the observed organomineralization, calcite nanofibers are of interest in the field of astrobiology and are considered as a potential sign of life. The moonmilk production contributed to rock formation by filling the pores and the cracks in the rocks, while the rock composition and the porosity shaped a microbial community that copes with high calcium content producing calcite nanofibers. The moonmilk is mainly found in karst caves, but there are examples of moonmilk bioprecipitation also in hypogeal environments carved in different geologic substrates, such as granitoid rocks or sandstone (Miller et al., 2018; Saladino et al., 2018). The moonmilk has also been found in lava tubes where the microbial communities are similar to those present in the moonmilk that originated from calcarenite (Gonzalez-Pimentel et al., 2021; Miller et al., 2020), raising the possibility of positing the moonmilk as a biosignature also beyond the Earth calcium carbonate rocks.

*Acknowledgements.* This work is dedicated to the memory of prof. Laura Frontali and the restorer Franco Adamo. The authors would like to thank Simone De Amicis for sharing sampling of the *macco* quarry of Tarquinia, Dr. Carlo Smriglio for the identification of the fossil showed in Suppl. Fig. 5, Domenico Mannetta for the carefully and professional preparation of thin sections and J.E. Hallsworth, for discussion and help while preparing the manuscript. We deeply thank Prof. Pierre Zalloua for manuscript revision and scientific advice. A.C. was awarded of the grant Regione Lazio PR FSE 2021-2027. This work was supported by Ateneo Sapienza 2021 (S. Ronca).

*Competing interests.* The authors declare that they have no conflict of interest.

*Author contributions.* S.R., M.B., A.C., F.B. and F.M. performed the experiments; M.B., S.R., T.R. collected samples; R.N., A.G. and M.R. supervised and performed the nucleic acid analysis; D.F.M supervised the sampling in the Etruscan tombs; T.R. and E.D.M. conceived the study and T.R. wrote the paper with contributions from S.R., M.B and E.D.M.

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

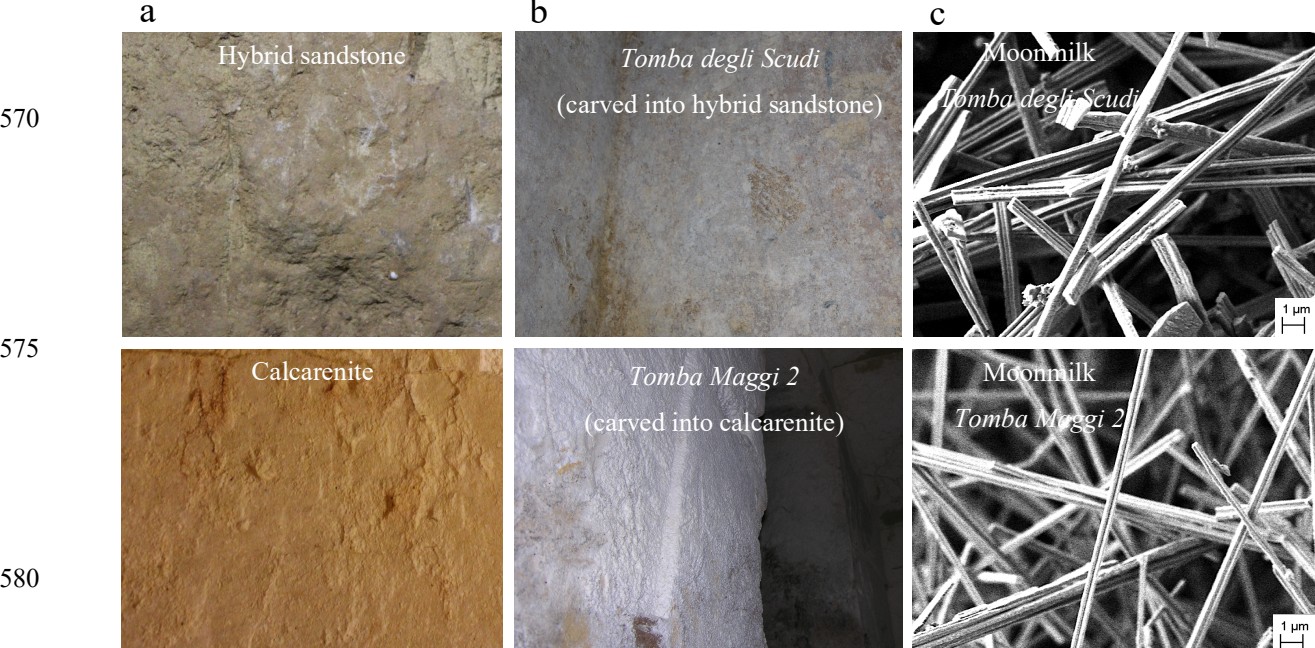

**Fig. 1. In Tarquinia, Italy, during the Iron Age, the ancient Etruscans carved hypogeal tombs into calcarenite and in hybrid sandstone bedrock whose walls are covered of the moonmilk, a secondary speleothem.** (a): examples of hypogeal walls of tombs carved in hybrid sandstone and calcarenite, the absence of the moonmilk is due to the restoration interventions. (b): along the centuries, the moonmilk speleothem precipitated as a white patina on the walls and ceilings of the tombs: as an example, the walls covered of moonmilk of the *Tomba degli Scudi* and the *Tomba Maggi 2* are shown, carved in hybrid sandstone and calcarenite, respectively. (c): scanning electron micrographs of the moonmilk sampled on the walls showed in (b) in the *Tomba degli Scudi* and the *Tomba Maggi 2*. Regardless the rock substrate in which the moonmilk is formed, the structure of the nanofibers is similar.

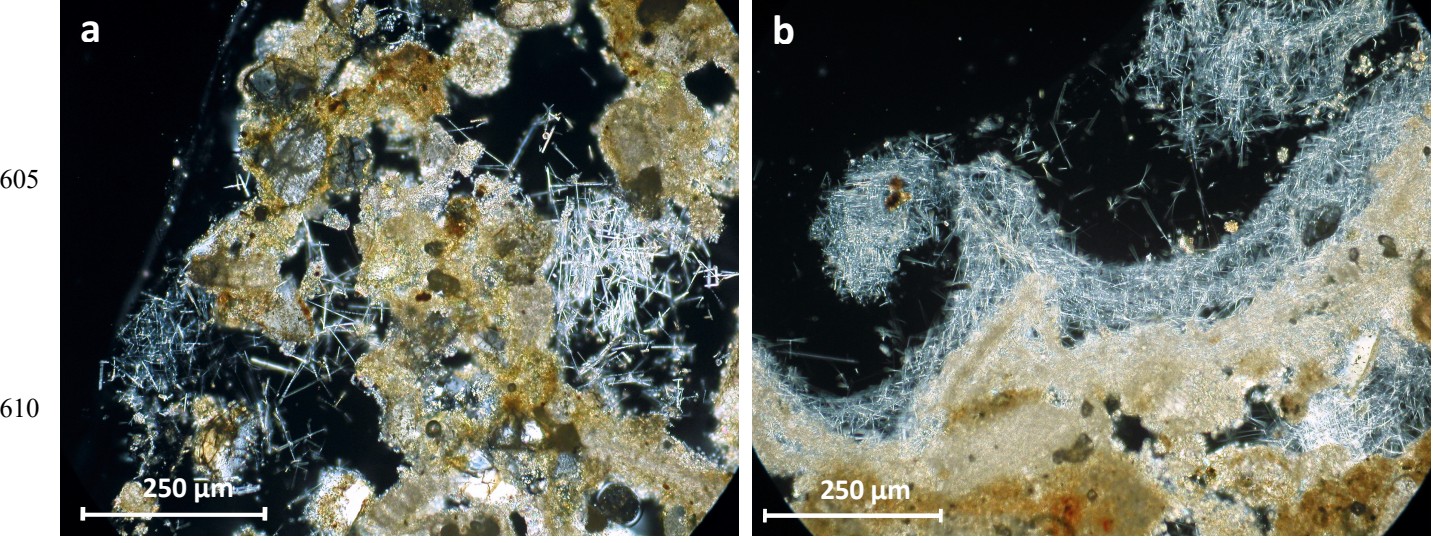

Fig. 2. The moonmilk is present on the surface and inside the calcarenite and sandstone rocks.

Optical microscope thin-section micrographs in crossed-polarized transmitted light. a: calcarenite sample collected inside the *Tomba Maggi 2;* b: hybrid sandstone sample collected inside the *Tomba degli Scudi.*

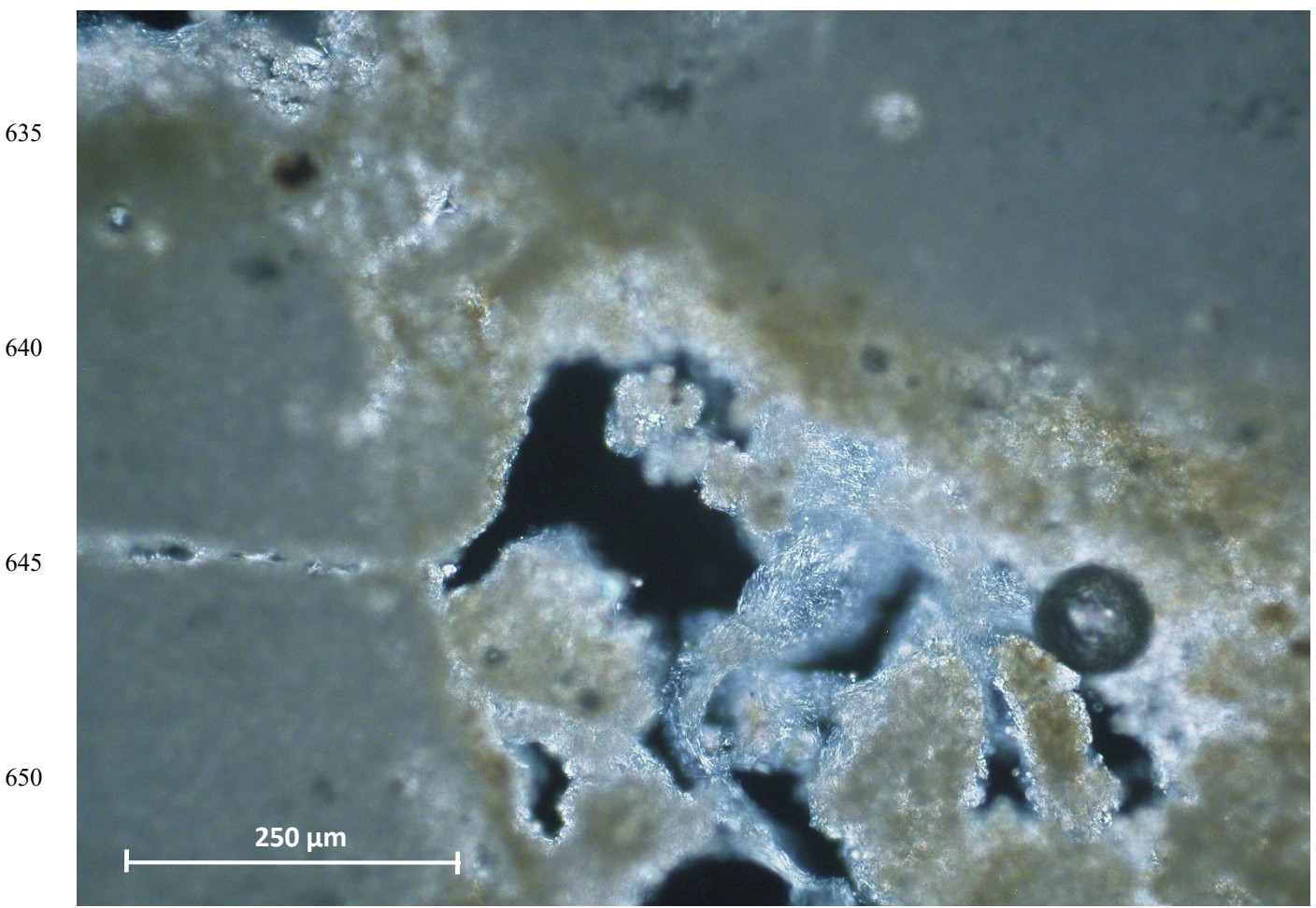

**Fig. 3. The moonmilk contributes to the lithogenic processes.** Optical microscope thin-section micrographs (crossed polarized transmitted light) of the moonmilk speleothems grown into the calcarenite sampled outdoor of *Tomba dei Vasi Dipinti*.

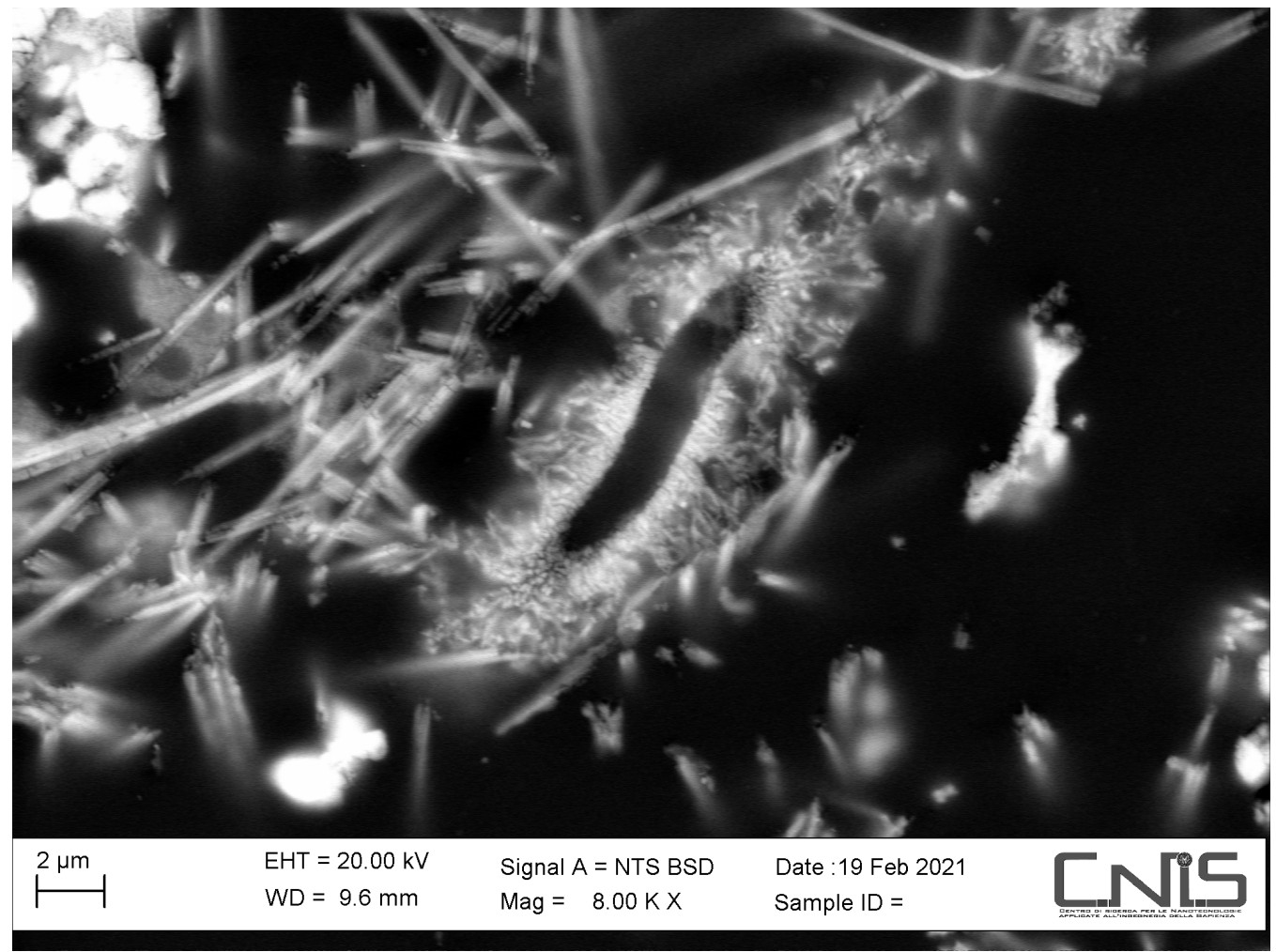

**Fig. 4. Bacterial biomineralization in the calcarenite.** *S*canning electron micrograph of a thin section of the calcarenite sampled outdoor of the *Tomba dei Vasi Dipinti*.

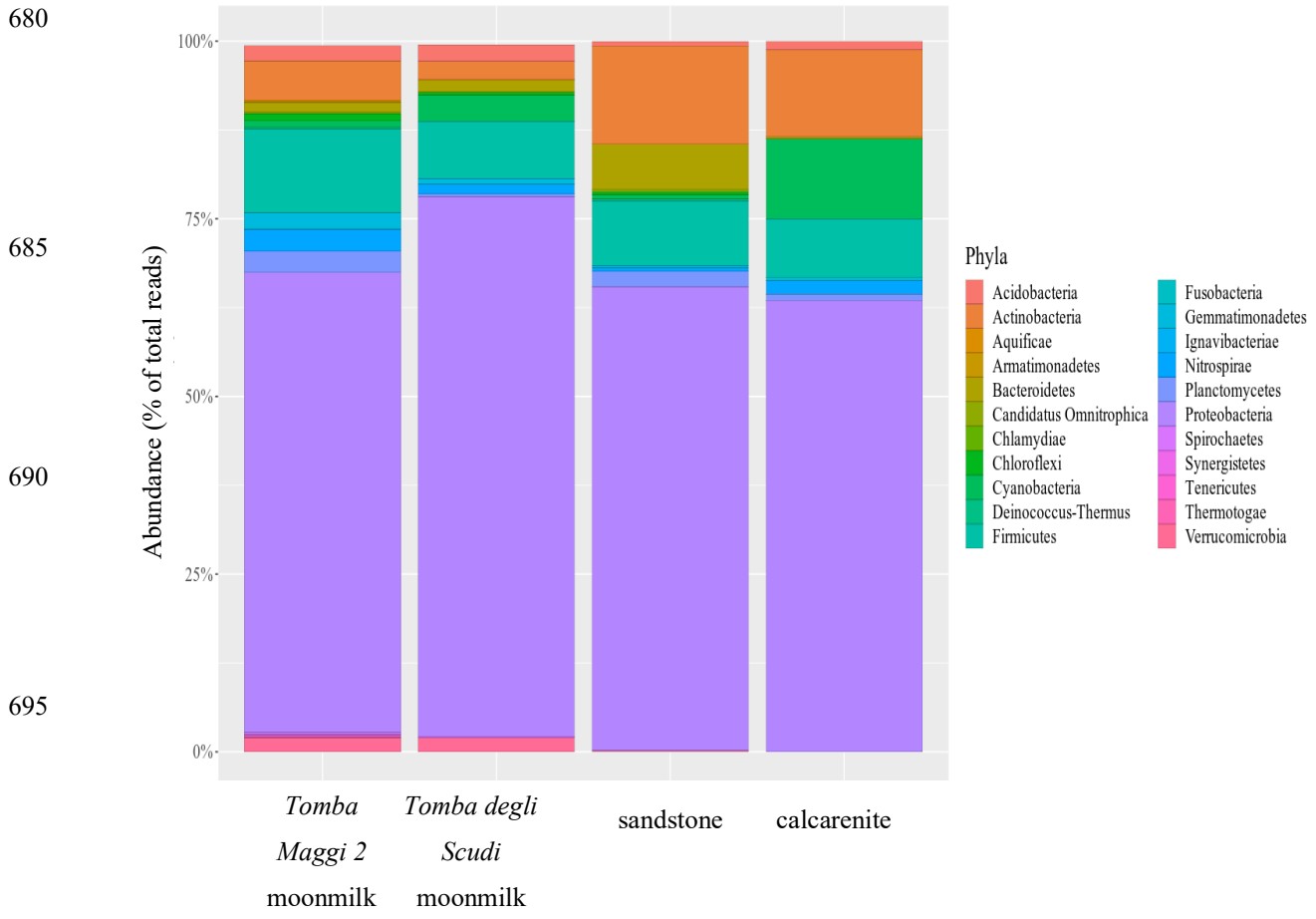

**Fig. 5. Phyla present in the microbial community from moonmilk samples of *Tomba Maggi 2, Tomba degli Scudi and* their corresponding rocks (calcarenite and hybrid sandstone). a:** The histogram shows the Phylum relative abundance (%) for the analysed samples. Community structure was determined by targeted amplicon sequencing of bacterial 16S rRNA genes. All samples show a high abundance of Actinobacteria, Bacteroidetes, Cyanobacteria, Firmicutes and Proteobacteria.