# Peer review of "Biogenic calcium carbonate as evidence for life"

_EGUsphere, 2022_

## Author Response (AR1)

Reply to Reviewer #1 comments **on the egusphere-2022-1457 manuscript**

*Dear Reviewer, thank you for your review and constructive suggestions on our manuscript "Biogenic calcium carbonate as evidence for life " (manuscript ID: **egusphere-2022-1457**). These comments are very valuable and helpful for us to revise and improve the manuscript. Below you will find answers to comments made in your reviews. The manuscript has been revised in accordance with your suggestions.*
*Yours sincerely*

This manuscript has an admirable goal- to determine whether or not certain forms of calcium carbonate constitute a biosignature. This is certainly of interest to the broad geobiologic and astrobiologic community (as outlined in the introduction), and I was intrigued by the sampling site and methods used. However there are some things that should be addressed before publication can proceed. Overall, I am very interested in the results of the paper, but framed such as they are currently, I find them inconclusive. Comments are as follows:

Lines 49-51: The Nutman et al. structures are highly controversial, the subject of great debate, and not at all accepted currently as definitive biogenic 'stromatolites'. They should not be presented so certainly.

*We thank the referee for this comment, the text was changed accordingly and adding the reference of Alwood et al 2006 (Stromatolite reef from the Early Archaean era of Australia).*

Lines 56-57: are the terms 'microbialite' and 'stromatolite' used interchangeably?

*Not exactly, we used microbialite to indicate carbonate rocks of microbial origin, while stromatolite follows the definition of Riding 2011." Stromatolites are macroscopically layered authigenic microbial sediments with or without interlayered abiogenic precipitates". Thus, we added "stromatolite (sensu Riding 2011)".*

Because the decline in stromatolite form diversity started in the Mesoproterozoic, with a large crash happening prior to the Neoproterozoic-Cambrian boundary, not in the Cambrian period (see papers by Frantz et al 2015; Awramik and Sprinkle, 1999).

*Thank you for this comment, we modified the text to clarify we are not talking about stromatolite form diversity, we are explaining that the microbial carbonate factory decreased because of the increase and the expansion of biologically controlled carbonate production operated by organisms which strictly regulate their calcification (sensu Pomar 2020). Thus, not only stromatolite, but in general the production of carbonate induced or influenced by microbes decreases because of the increase of organisms with carbonate skeletons.*

Also, many of the Archean forms are dubious in their biogenicity (Grotzinger and Knoll, 1999; Mclauglin et al, 2007).

*Thank you for the comment; however, biogenic stromatolites are abundantly recorded (Alwood et al 2006, 2009, 2014, Schopfs 2011).*

This seems to be my main general problem with the manuscript- the assumption is that all carbonates are biogenic instead of acknowledging that these forms are ultimately ambiguous.

*In line 45 we wrote "Carbonate rocks on Earth are of abiogenic or biogenic origin". In the following lines 60-87 we discussed this point. We thank the referee for raising this point and the text was changed. However, recently, increasing evidence was found that many abiotic carbonates are contrarily of biogenic origin. For example, the ooid carbonate grains were considered for long time of abiotic origin. Recently Diaz and Eberli (2019) demonstrated that ooids are of microbial origin, considerably reducing the fraction abiotic carbonate production.*

Certainly, if we are saying that carbonates on Mars mean that there was once life on Mars, the drive is on us to prove beyond a shadow of a doubt that the structures here on Earth are/were all biogenetic. It cannot just be assumed that they are.

*Thank you for the comment, we clarified the text to express a different concept: (line 77-80= Carbonate rocks, being mostly of biogenic origin, could be considered in this perspective, but even if specific analyses could discriminate between biotic and abiotic carbonate rocks (Blanco et al., 2013), the attempt to unequivocally distinguish between carbonates of biogenic or abiogenic origin remains vacuous, especially in a search for evidence of life on other planets). Our only aim is to provide evidence that the formation of calcium carbonate nanofibers are biologically controlled/driven and, potentially, could represent a biosignature.*

Lines 77-78: same problem as above- has it been conclusively proven that most carbonates have biogenic influence? The null hypotheses should be that these structures are abiogenic. Without that, the logic ends up being circular, and it is argued that these structures must be biogenic because carbonate structures are biogenic.

*As above, in the line 75-80 we are explaining that "the mineral phases resulting from abiogenic or biogenic activity are indistinguishable, and the identification of irrefutable biosignature, evidence of past or present life, is still lacking. The attempt to unequivocally distinguish between carbonates of biogenic or abiogenic origin remains vacuous".*
*We are not concluding that carbonates are only biogenic, and, we apology if the text gave the idea we wanted to conclude that carbonate are only biogenic, we have now clarified this point.*

Line 86: The authors rightly acknowledge that direct evidence for biogenicity is lacking. This should be the main point. Many research *assume* biogenicity, without direct evidence. The utility of this paper is an attempt to provide a direct link. Reframing the subject around this point would be more impactful.

*Thank you for this comment, we have updated the text accordingly. Focusing on the bio related specific CaCO₃ structure, and not the CaCO₃ itself should solve this point.*

Lines 93- 95: I am not familiar with the previous work on the subject, but why does containing an active microbial population mean the forms are conclusively biogenic? Microbes may just be living in the structure. Just because I live in a building does not mean I built it (see arguments in Grotzinger and Knoll, 1999, Petryshyn et al. 2021). It is nearly impossible to find a surface on Earth that does not contain an active microbial community.

*We totally agree on this point, and we have clarified the text to report that in this case, the absence of karst features, such as abundant dissolution and reprecipitation processes with development of carbonate concretions and meteoric cements, is indicative that inorganic calcium carbonate precipitation from interstitial meteoric waters does not occur in the bedrock*

*where the tombs are carved. Moreover, the area of interest in this study insists on a high flat relief, in which the infiltrated water mostly derives from rainfall, without any groundwater circulation. Such evidence, along with the diffuse presence of microorganisms in the carbonate nanofibers deposit supports that the moonmilk origin could be controlled or driven by microbial metabolism. Finally, the Phila already analysed in other karst caves all over the world are very similar to the Phila reported from our studies, and this study could suggest a possible natural selection of specific microorganisms adapted to rocks rich in calcium carbonate.*

Line 94: is there any mechanism known by which abiogenic calcite nanofibers are precipitated?

*We have added in the text a paragraph concerning hypothesis on the origin of the fibers, as an example, a paper was published claiming the abiogenic origin of calcite fibers: Jones, B., & Peng, X. (2014). This paper states: "that the calcite fibers found in the spring deposits at Shiqiang were a late diagenetic product that **probably** formed through abiogenic precipitation". In this case evidence is reported that spring waters led to the formation of the fibers. By the way, there is no demonstration that these fibers are not biogenic, there are not microbiological experiments in this paper but only SEM analysis. Indeed, in fig 14F an entombment is showed, and many microorganisms are also present in SEM pictures. It should also be noted that the SEM analysis is not suitable for microbe observations because the largest part of microorganisms is destroyed by the vacuum of the SEM chamber. That's the reason why only filamentous microorganisms such as fungi and actinomyces or their spores are frequently observed; below an example is Figure 3, Moonmilk deposit in the Etruscan Stanza degli Scudi (Cirigliano et al 2018).*

Methods are good, sound, and well-described. I don't see a 16S plot of results in the figures or supplemental files. This would be interesting to look at.

*Experimental 16S data have been added to the supplementary materials.*

Lines 182-195: It is very interesting that the moonmilk is found within the walls as well. Is it possible that it was deposited there previously, and then covered by a new layer? Or is it possible that endolithic bacteria have burrowed into the wall, causing retrograde neomorphism and creating void space that could later be filled?

*Data describe the extant microbial community and do not provide information about the previous history of the stones. Concerning the walls of the tombs, we have already published observations about the thickness of the moonmilk patina, comparing different tombs. Thera are differences in moonmilk thickness from calcarenite (thicker) and sandstone (thinner). This result correlates with higher calcium content in calcarenite than in sandstone. The interpretation is that microorganisms in calcarenite must cope with higher calcium content potentially detrimental for them, inducing a thick monnmilk deposition (Cirigliano et al 2021a). Thus, in this manuscript we present, for the first time, a comparison of the endolytic community of the rocks with the respective moonmilk community. We show in the manuscript that they are statistically very similar, hence the importance to deeply investigate in the future the microbial-rock interaction.*

A scan of the 16S data might reveal the presence of endoliths, and would be helpful in interpretation of the SEM and petrographic analysis.

*In this work we show that both the internal (endolytic) and the moonmilk forming external microbial community are statistically the same. Macroscopic and microcroscopic observations along with SEM analysis revealed that the moonmilk concretions developed inside the calcarenite in the vuggy and moldic porosity and in intergranular and moldic pores of the hybrid sandstone bedrock (Fig. 2., 3 and Supplement Fig 2-4), (Fig. 2b), regardless of where the samples were collected, outdoor or indoor. As shown in Supplem. Fig 5, moonmilk was also found in the void space at the contact between a fossil bivalve and the host calcarenite (Supplementary Fig. 5). Although dissolution effect induced by bacterial communities on the carbonate rock substrate cannot be excluded, the high porosity of the rocks is surely a peculiar character of the rocks hosting the moonmilk and, it is the physical chemical features of the substrate (i.e. the high value of open porosity and the carbonate composition) that promote the microbial colonization and the moonmilk deposition.*

Line 208-214: This point, and SI Figure 9, are incredibly intriguing. However, with the evidence given, it may just be that the urea in solution fostered the precipitation of the calcium carbonate, as it is known to do, and that this nucleation centered around the colony because the charged surface is a good template for precipitation. Ideally, in order for this to be conclusive, I would like the authors to consider a labeled bicarbonate uptake experiment. If the labeled bicarbonate is incorporated into the calcite crystals, it would conclusively show that microbial metabolism is driving precipitation. Otherwise, this remains intriguing but inconclusive.

*Thank you for the comment, the experiment with labeled bicarbonate will be considered for future experiments, by the way, since moonmilk cannot be reproduced in laboratory conditions, again this will be an indirect evidence. The result presented in supplementary 9 reports a carbonatogenic strain able to activate metabolic pathways promoting calcium carbonate precipitation. We tested hundreds of bacterial strains in the laboratory, and only a small percentage promotes calcium carbonate precipitation. Thus, many bacterial colonies just grow on YPD, urea and $CaCl_2$ without promoting calcium carbonate precipitation. Our evidence is that bacteria are centers of nucleation for nanofibers formation, they do not control the mineralization process directly, but induce the precipitation of calcium carbonate by changing the chemistry of the environment as a consequence of their metabolic activity and also serving as nuclei for crystallization. The mechanism consists of pH increase, direct effect of the negatively charged bacterial surface, presence of a metabolic ureolytic process. The three causes are intimately connected and are not untangled in the present analysis. Finally, our data and published data (Banerjee, S., & Joshi, S. R. (2014). Ultrastructural analysis of calcite crystal patterns formed by biofilm bacteria associated with cave speleothems. Journal of Microscopy and Ultrastructure, 2(4), 217-223.), showed that inactivated (killed) cells were unable to precipitate CaCO3 in laboratory, suggesting that cells need to be metabolically active for calcification and that cell structure alone is not sufficient to promote bioprecipitation.*

Line 226: This should be noted as 16S SSU rRNA analysis, not 16S RNA.

*Thank you, the text was changed as suggested.*

Line 230-235: I agree with the authors, no habitat should be considered 'extreme' or a 'refuge'- microbes like it there!

*Thank you for this comment, we consider this point very important.*

Comment on egusphere-2022-1457', Anonymous Referee #2,

*Dear Reviewer, thank you for your review and constructive suggestions on our manuscript "Biogenic calcium carbonate as evidence for life " (manuscript ID: egusphere-2022-1457). These comments are very valuable and helpful for us to revise and improve the manuscript. Below you will find answers to comments made in your reviews. The manuscript has been revised in accordance with your suggestions.*
*Yours sincerely*

General Comments:

The manuscript has two goals: 1) to provide the first description of moonmilk and associated microfossils within rock cavities, and 2) to provide evidence for the biogenicity of moonmilk.

Both goals fit within BG's scope. The first goal is attained well, with certain areas that could be opened up for further discussion noted in the Specific Comments below. However, the second, larger goal to assess moonmilk biogenicity does not have enough evidence to support the paper's conclusions.

The first argument for biogenicity discusses microbial cells entombed in moonmilk crystals, as shown in Fig. 4 and Sup. Figs. 6-9. The pictures are beautiful, but do not conclusively prove that microbial metabolisms were the source of crystallization. In short, mineral precipitation on microbial cells can either occur 1) as microbial metabolisms influence local chemistry, or 2) due to external abiotic chemical conditions, just like a stick becomes encrusted with minerals in a hot spring (ex: Dupraz et al., 2009, Petryshyn et al., 2021, Geobiology).

*Thanks for comment. The reviewer quoted some examples (Dupraz et al., 2009, Petryshyn et al., 2021) in which it is clear that the physio-chemical process for abiogenic carbonate precipitation is clearly driven by $CO_2$ degassing. In the Petryshyn case history, there is a spring, and the abiotic calcium carbonate precipitation takes place close to it. On the contrary, the investigated moonmilk developed in caves and rocks which are located in a flat elevated area, in the shallow vadose zone in which infiltrated water mostly derives from rainfall. Evidence of water circulation is generally lacking. On the other hand, if it was present, it should have left a diagenetic footprint producing meteoric mosaic cement between pores with clear (limpid) inclusion-free crystals.*

For example, silica frequently precipitates on microbial cells in supersaturated conditions without any influence from metabolisms (Benning et al., 2004, GCA). The entombed cells are biosignatures, but the silica itself is not. Similarly, the microbial cells entombed in moonmilk are biosignatures, but unless the manuscript can show that moonmilk only occurs in the presence of biology, moonmilk itself cannot be a biosignature.

*Thank you for this comment, indeed, to produce moonmilk, not only the negatively charged bacterial surface is needed, but also the specific metabolic pathway that is activated to cope with high calcium content is required. The biogenic $CaCO_3$ precipitation is widely studied for several technological applications (as an example, we studied this phenomenon in laboratory for bioconsolidation in the field of Cultural Heritage, but it is studied also for self-healing cement, sand consolidation etc.). We have not mentioned this point in the text to avoid confusion to the readers, here we are reporting a natural phenomenon. Thus, the mechanism of biogenic precipitation is well studied, the most common (and fast) metabolism for $CaCO_3$ precipitation is the ureolytic metabolic pathway. This process involves the production of carbamate $(NH_2COOH)$ by urea hydrolysis, which spontaneously hydrolyses to form ammonia $(NH_3)$ and carbonic acid $(H_2CO_3)$. These products react with water to form carbonate $(CO_3^2)$, ammonium ions $(NH_4^+)$ and hydroxyl ions $(OH^-)$, finally resulting in an increase of pH. In an alkaline environment, the presence of calcium ions and the bacterial cells, as nucleation site, allows the precipitation of calcium carbonate. Thus, it is not the presence of the bacterium alone, as a structure, that provides evidence of precipitation, but also high pH, metabolism, and negative charged membranes. In laboratory conditions (suppl Fig. 9), if urea is present, the mechanism is stimulated/accelerated.*

The second argument for biogenicity involves the similarity of microbial communities within rocks and on their surfaces. While interesting, similarities between the two communities is not enough evidence to support biogenicity. For example, silica precipitation can preserve the same cyanobacteria on sinter surfaces and interiors (Phoenix et al., 2006, Geobiology). Supersaturated environments, whether hydrothermal or cavern fluids, will entomb similar microbes equally regardless of location.

*In the area we studied, there is no evidence of cavern fluids or hydrothermal fluids in the investigated tombs. Actually, the absence of karst features such as abundant dissolution and reprecipitation processes with development of carbonate concretions and meteoric cements indicates that inorganic calcium carbonate precipitation from infiltrated meteoric waters does not occurs in the bedrock where the tombs are carved.*

In short, the manuscript describes an interesting new location to search for microfossils and unique mineral precipitation, but the discussion of biogenicity either needs to be removed, or supported by new datasets.

*We hope the answers to comments above clarified the point that not only the physical presence of microorganisms promotes the nanofibers formation, but also the active metabolism. Of course, nobody knows how these nanofibers elongated (Cirigliano et al 2018), and this point deserves more scientific research in the future and it is not the goal of this paper. In the future, the unique possibility to compare the moonmilk from 13 tombs could provide inspiration for future experiments.*

I agree with the previous reviewer that labeled carbon isotope experiments would be a great way to test if metabolisms truly cause carbonate precipitation (e.g., Wilmeth et al., 2018, Front. In Microb.)- however, these methods are potentially too costly and/or time intensive for the scope of this study. Another alternative dataset would be to show that groundwater inside and around the tomb is undersaturated with respect to calcium carbonate, and could only precipitate with the assistance of microbial communities.

*Thank you for the comments, we will continue this research and we will take this point into account.*

Specific Comments:

Lines 37, 45, 66, 75-81: Like the previous reviewer, I agree that the introduction should shift away from the idea that carbonates are usually biogenic, especially since several types of abiotic carbonates are mentioned. Instead, the authors can make a simple change throughout the introduction: "many biogenic deposits are made from carbonate minerals". There is a subtle difference between "carbonate minerals are usually biogenic" and "many biogenic deposits are made from carbonate minerals". I think the second idea is closer to the point the authors make in Lines 77-81, that it's difficult to tell biogenic from abiogenic, which I definitely agree with.

*Thank you, the text was changed in some parts as suggested.*

Lines 49-51: The Nutman samples are highly controversial and can be removed from the discussion without damaging the arguments made.

*This part of text was changed and the reference to Nutman et al 2016 structures was removed.*

Lines 82-93: This specific section on moonmilk should be expanded to discuss research history and the debate of biogenic vs. abiotic formation. For example, Borsato et al., 2000 concludes that moonmilk is abiotic, an idea which is not mentioned in the current manuscript. At this stage, the reader would conclude that all previous moonmilk literature assumes biogenicity. A more detailed discussion would provide more context for the evidence presented.

*Thank you for the comment, the text was revised. The paper Borsato et al., 2000 have already been cited, it is an important paper. Indeed, in their SEM images there are a lot of structures (figs 4-6) that could be microbial calcifications (a microanalysis is not presented in the paper, it is impossible to know if the fibers are calcium carbonate) and the authors stated in fig5: "These filaments are commonly composed of nanofibers, micrite grains, and biogenic (?) filamentous material.". Thus, there is not a clear demonstration of a-biogenicity origin.*

Lines 96-97, 216: I'm not sure what is meant by a "geological process", can you please be more specific?

*Thank you, we changed this part of the text for clarification.*

I have no issues with the methods and materials- this is a very interesting sampling location, worth further research!

*Thank you for the comment, indeed we are asking permission to take new samples to perform new experiments (also following the comments to this manuscript) and to get the physical parameters inside and outside the tombs for a long period of time.*

Lines 183-199: Since this is the first description of moonmilk within rocks, a broader comparison with other moonmilk textures would be very helpful. Are there any similarities or differences between moonmilk inside the rocks vs. outside, or in other moonmilk locations?

*Macroscopic and microscopic observations along with SEM analysis revealed that the nanofibers have developed inside the calcarenite in the vuggy and moldic porosity and in intergranular and moldic pores of the hybrid sandstone bedrock (Fig. 2., 3 and Supplement Fig 2-4), (Fig. 2b), regardless the samples were collected outdoors or indoors. As shown in Supplem. Fig 5, the calcium carbonate nanofibers was also found in the void space at the contact between a fossil bivalve and the host calcarenite (Supplementary Fig. 5). Although dissolution effect induced by bacterial communities on the carbonate rock substrate cannot be excluded, the high porosity of the rocks is surely a peculiar character of the rocks hosting the moonmilk and also the physical chemical features of the substrate (i.e. the high value of open porosity and the carbonate composition) that promote the microbial colonization and the moonmilk formation.*

Lines 204-214: See General Comments. Entombment alone is not enough to prove biogenicity-minerals can precipitate on cells without direct metabolic influence. The manuscript needs to show that moonmilk can only be made in the presence of microbes.

*From a geological point of view, the entombment due to inorganic precipitation of calcium carbonate may occur in environments with circulation of supersaturated fluids (i.e springs, hot springs, cascades, lacustrine areas). The area analysed here corresponds to a high flat relief, in which the infiltrated water mostly derives from rainfall, without any groundwater input.*

*From a biological point of view, by now, it is impossible to reproduce moonmilk in laboratory conditions. By the way, we can report that we have experience in calcium carbonate production in the laboratory, both from pure bacterial culture (an example is in suppl fig. 9) and from rock microbial cultures. We have evidence that the grinded calcarenite, with its entire microbial community, in a medium containing urea and CaCl2 easily and fast produced pure calcite. On the contrary, pure carbonatogenic bacterial cultures produce vaterite (Benedetti et al. 2023 submitted to J. of Cultural Heritage). It's clear that the field of microbial-rock interaction studies is still in its infancy. Of course, as already highlighted microbes are everywhere, by the way, all over the world, moonmilk always hosts very similar microbial communities that again, indirectly, suggest the intimate link microbes-rocks. This is the logical limit of the concept of the moonmilk biogenesis and the limit of its potential value as a beacon for life.*

Lines 210-214: If there have been any hypotheses regarding how microbes produce nanofibers, even if currently untested, they should be discussed here. I agree with the authors that precipitation experiments are best performed using the entire microbial community and not single cultures alone. If it's possible to perform labeled isotope experiments or moonmilk saturation experiments on biofilms, that would be very interesting data. See also lines 82-93-alternative hypotheses regarding abiotic moonmilk formation should be discussed here as well.

*Thank you, we clarified these points in the introduction. As stated above, we have evidence that the grinded calcarenite, with its entire microbial community, in a medium containing urea and CaCl2 easily and fast produced pure calcite. On the contrary, pure carbonatogenic bacterial cultures produce vaterite (Benedetti et al. 2023 submitted to J. of Cultural Heritage). Suggestions will be taken into account because we are planning experiments to try to understand the origin of nanofibres, a very complex point, that is beyond the scope of this paper.*

---

## Referee Report (RR1)

The authors have addressed many of the questions from the previous round of reviews. Overall, I think they have begun to strengthen their case for the biogenicity of moonmilk in Tarquinia, but there are still three major areas of revision that need to happen before publication. In short, the argument for biogenicity is divided into three parts- environment, cultures, and RNA.

First, the authors note that meteoric waters in the tombs should be undersaturated with respect to calcite (Section 3.1), generally fostering dissolution. I broadly agree with this point, but it should be noted that meteoric waters dissolving calcite will gradually increase in saturation, re-precipitating crystals elsewhere without the need for life. The authors discuss these potentials, but the text contains several contradicting statements about the presence/absence of carbonate dissolution and abiotic precipitation of cements, noted in detail below.

Second, the text compares culturing experiments between living and sterilized bedrock- this is the strongest evidence for biogenic moonmilk in the paper. However, as noted below, this argument relies on data from an unpublished paper. Furthermore, the culturing experiments from this research are missing from the Methods section, and need an expanded discussion to compare lab conditions with in-situ environments.

Finally, the text asserts that similar microbial communities in bedrock and moonmilk is a sign of biogenicity. In the detailed notes below, I repeat the argument that there are many abiotic scenarios where similar microbial communities can be preserved in different parts of the same rock. In short, the similarity alone cannot be a biosignature.

**1) Section 3.1**
In my first comments, I mentioned that minerals can entomb cells without requiring microbial metabolisms, especially in supersaturated environments. The authors responded that the tombs are in the vadose zone, that there's no carbonate-rich groundwater source nearby, and that any water in the system is likely meteoric. I agree with the authors on all these points. The authors added these points in Lines 265-280, and they strengthen the manuscript.

However, the processes of carbonate dissolution and precipitation in the tomb walls need to be clarified. At the moment, there are a few contradicting statements about dissolution and precipitation in the bedrock. For example, in Lines 249-250 describing Macco facies,

> "*In the inner walls of the intergranular voids* **microsparite cement precipitation and/or recrystallization** *often occurs, due to* **diagenetic processes of dissolution**.*"

This sounds reasonable to me- the vadose zone often shows signs of dissolution followed by precipitation, as previously undersaturated waters become concentrated in Ca and CO3. However, in Lines 267-9 describing the general bedrock:

> "*Noteworthily, the characteristics of bedrock porosity indicate that the* **dissolution processes prevail** *on those of* **inorganic carbonate precipitation**; *indeed,* **meteoric cements** *as well as speleothems in the largest cavities* **are absent.** *Moreover, the bedrock where the tombs are carved is located in the shallow vadose zone (few meters below the surface) and* **it is barely susceptible to dissolution-reprecipitation processes,…**"

In short, Section 3.1 says that the bedrock 1) contains evidence for dissolution and cement precipitation, 2) mostly evidence for dissolution with no cement precipitation, and 3) barely any dissolution or precipitation. These contradictions either need to be corrected, or if the authors are talking about different locations, that information needs to be clarified.

Here's why the discussion of dissolution and precipitation is important. A convincing argument for biogenic moonmilk requires: 1) a source of calcium, provided from calcite dissolution somewhere in the vadose zone, and 2) undersaturated meteoric waters which are less likely to precipitate calcite, requiring microbial metabolisms to foster precipitation. The authors address these points (end of 3.1), but there are two major questions left to address:

If the bedrock has no evidence for calcite dissolution, where is the calcium coming from to fuel biogenic moonmilk formation? On the other hand, if abiotic cements are present elsewhere in the bedrock, indicating periods of supersaturation, why does moonmilk need to be biogenic?

For publication to continue, the authors should:

> 1: clarify their interpretations on patterns of precipitation and dissolution in the bedrock- is it an environment that contains both dissolution and precip? Is it dominated by dissolution? Or are both dissolution and precip limited?
>
> 2: once the authors have a consistent interpretation of precip/dissolution, clearly describe how such groundwaters create conditions where only microbes can make moonmilk (the authors start to do this in Lines 278-279).
>
> 3: (Optional) If the authors have any data on local groundwater, especially pH, Ca, and estimates for calcite saturation, that would greatly help build their case for a subsurface environment where only microbes could make moonmilk. This data could be from the authors themselves, or collected from the literature.

2) Section 3.3- cultures
The strongest evidence for biogenic moonmilk at Tarquinia is presented in Line 329:

> *"Indeed, under laboratory conditions, we have evidence that the grinded calcarenite, with its entire microbial community, when present in a medium containing urea and CaCl2 produced calcite. **In the same conditions, calcium carbonate is not produced with sterile (autoclaved) grinded calcarenite (Benedetti et al. 2023)."***

This is exactly the type of experiment needed to show the biogenicity of carbonate precipitation. However, I have a few notes about the culture experiments and references.

First, Benedetti et al., 2023 is still in review. I don't know Biogeosciences' citation policies, but many journals require such references to be cited as (Benedetti et al. in review). More importantly, if this crucial data is rejected by the scientific community, then the argument for moonmilk biogenicity in this paper becomes much weaker.

This paper provides its' own culture experiments, as mentioned in Lines 304-329 and shown in Supp. Fig. 9. The methods for these experiments must be described in the Methods section- my apologies for missing that note last time. How much CaCl2 was used? How much urea? What was the temperature and carbonate saturation of the experiments? Most importantly, were sterilized experiments run during the research for this paper (not the unpublished work)? If so, sterilized comparisons would greatly help convince an audience that moonmilk is biogenic.

The paper also needs to describe differences between the culturing experiments and the natural conditions in the bedrock. While it is impossible to exactly replicate the tomb environment inside a culture, the differences must still be addressed in the discussion. For example- do the authors think that Ca and urea are abundant in the bedrock environment? Especially important: What was the saturation of calcite in these experiments, and how does that relate to potential saturation states in the bedrock?

On a similar note, the authors provide a nice description of urea hydrolysis in the author reply. They mentioned leaving it out of the manuscript to avoid confusion, but I think this metabolism should be included in the discussion. Otherwise, the reader is missing crucial information on a potential process behind moonmilk formation.

For publication to continue, the authors should:
      1: Describe culturing experiments in the Methods.

      2: Change Benedetti references to "in review".

      3: Provide extra discussion on the differences between culturing experiment conditions and conditions in the tombs- especially for chemical conditions (temperature, Ca, urea, calcite saturation).

      4: Include a description of urea hydrolysis in the discussion. The best location is probably alongside the discussion of culturing experiments.

3) Section 3.3- RNA
The paper asserts that similar 16S RNA in bedrock and moonmilk are evidence for moonmilk biogenicity. While the results are interesting, and belong in the paper, such similarities alone are not enough to determine biogenicity, for one main reason.

There are other scenarios where moonmilk and bedrock communities are similar, but do not require biogenic moonmilk. For example, if the same microbes inhabited the entire porous bedrock sample before moonmilk formation, and then moonmilk formed abiotically through environmental change, both locations should have the same microbial populations. Therefore, similar populations can not distinguish biogenic vs abiogenic moonmilk.

When I brought this idea up in the previous round of reviews, the authors responded that the environment was unlikely to precipitate calcite abiotically. However, my issue here is not with

environmental saturation- I've already addressed that topic. My issue is with the claim that similar RNA data can be used as evidence for biogenic moonmilk, as stated in Line 346:

> *These results show that **in moonmilk and rocks the microbial composition is similar,** irrespective of rock type (calcarenite and hybrid sandstone) or the environment where the samples were collected (outdoor or indoor). It should be noted that 16S SSU rRNA analysis does not provide information about metabolic activity, thus these data do not identify microorganisms that are active in $CaCO_3$ deposition, but **the overall data demonstrate that the endolytic community of the rocks is promoting moonmilk deposition.***

In short, if the response to the question "Are similar communities in moonmilk and bedrock a biosignature?" is "Yes, but only because the environment cannot precipitate moonmilk", then the biosignature is not the RNA itself, but the presence of moonmilk in an undersaturated environment.

For publication to continue, the authors should:

> 1: Keep the RNA data and keep the comparisons of similar communities in the bedrock and moonmilk. However, any sentences that mention the RNA itself as evidence for biogenic moonmilk need to be removed (Lines 338-339, 350-352).

---

## Author Response (AR2)

Author's response to the referees
Reviever 1

**Suggestions for revision or reasons for rejection**
(visible to the public if the article is accepted and published)

This revised manuscript investigates the formation of calcite nanofibers (moonmilk) in the Monterozzi Necropolis in Central Italy. It is an intriguing field site with beautiful microfabrics. I appreciate the large amount of work the authors did in order to address reviewer comments on a previous version. The topic of the biogenicity of carbonates is of wide interest to geo/astrobiologists, and the complexities of the issues are well stated in the introduction. There are a few issues here and there (outlined below), but after minor revision, I believe this manuscript is suitable for publication.

For instance, the term 'bacterial biomineralization' or just 'biomineralization' is used throughout the paper, especially in section 3.3, enough so that a definition of the term would be useful, since I assume the authors do not mean direct biomineralization in the way shells and bones are produced, but more of indirect biomineralization wherein microbial metabolisms influence the surrounding geochemistry and foster precipitation. Dupraz et al. 2009 (Earth Science Reviews) defined 'organomineralization' for this class of precipitation. Overall, a definition of what the authors mean by the term is necessary up front (or they can go with the Dupraz classification). *Thank you for this comment, we added the term Organomineralization sensu Dupraz et al 2009*

My biggest concern is the treatment of the laboratory results that being by line 304. It is unclear to me what 'in plates' refers to- is this is reference solely to the previous work, or did the authors do some of their own experiments? If their own, the methodology should be included. In line 325 it seems this might be referencing another paper- however I am not sure. Since it is a good line of evidence for microbial involvement in precipitation, which is the main goal of the manuscript, it would be useful if the experimental conditions were described in a bit more detail.

*Thank you for this comment, the same point has been raised by the second referee, we have performed a new experiment, not present in the paper under review, in supplementary fig 10, showing that the microbial community of the calcarenite precipitates high amount of calcium carbonate and we updated the Methods section.*

Line 44: might be better to state that carbonates form in/from environments most likely to contain life, rather than assuming biogenicity.
Line 57: use "biogenic" rather than 'biotic', since organisms are not directly making the carbonate (eg, through biomineralization)
*Thank you, we changed biotic with biogenic in the text.*

Lines 330-335: a breakdown of the specific metabolic processes happening here would be useful. What is the overall metabolism producing a rise in pH? Is the rise in pH actually due to bicarbonate/carbonate production, and how is the presumed saturation index impacted by the metabolism? How is the presence of a metabolic ureolytic process confirmed? Phylum level diversity won't necessarily shine any light on whether or not he microbes in the sample are capable of these metabolic processes, but there are a high proportion of Firmicutes (or Bacillota), which are known to contain several members (such as those in Lysinibacillus groups) that have been shown to have extremely high urease activity and therefore greatly enhance carbonate precipitation. It should be pointed out that this phylum is present and fairly abundant.

*we update the text following these suggestions, we have also detailed that the strain of the supplementary figure 9 is indeed a Lysinbacillus species.*

The pictures are beautiful, especially in the supplemental files. The fibers entombing bacteria – wonderful!
*Thank you!*

**Report #2**

Submitted on 23 Jun 2023
Anonymous referee #2

**Suggestions for revision or reasons for rejection**
(visible to the public if the article is accepted and published)

(modified from Response to Authors)
The authors have addressed many of the questions from the previous round of reviews.Overall, I think the authors have begun to strengthen their case for the biogenicity of moonmilk in Tarquinia, but there are still three major areas of revision that need to happen before publication. In short, the argument for biogenicity is divided into three parts- environment, cultures, and RNA.

First, the authors note that meteoric waters in the tombs should be undersaturated with respect to calcite, generally fostering dissolution. I broadly agree with this point, but it should be noted that meteoric waters dissolving calcite will gradually increase in saturation, re-precipitating crystals elsewhere without the need for life. The authors discuss these potentials, but the text contains several contradicting statements about the presence/absence of carbonate dissolution and abiotic precipitation of cements.

*Thank you, we fixed this contradiction in the text. We showed that the dissolution process is dominant and can make available $Ca^{2+}$ for microbial metabolism.*

Second, the text compares culturing experiments between living and sterilized bedrock- this is the strongest evidence for biogenic moonmilk in the paper. However, this argument relies on data from an unpublished paper. Furthermore, the culturing experiments from this research are missing from the Methods section, and need an expanded discussion to compare lab conditions with in-situ environments.

*The referee is right, a new experiment, never published before and not present in the paper under review, has been added, in supplementary fig 10, showing that the microbial community of the calcarenite promotes the precipitation of high amount of calcium carbonate. When the calcarenite is sterilized, there is no calcium carbonate production. We also updated the Methods section.*

Finally, the text asserts that similar microbial communities in bedrock and moonmilk is a sign of biogenicity. There are many abiotic scenarios where similar microbial communities can be preserved in different parts of the same rock. In short, the similarity alone cannot be a biosignature.
*Following also the comments below, we modified the text.*

1) Section 3.1
In my first comments, I mentioned that minerals can entomb cells without requiring microbial metabolisms, especially in supersaturated environments. The authors responded that the tombs are in the vadose zone, that there's no carbonate-rich groundwater source nearby, and that any water in the system is likely meteoric. I agree with the authors on all these points. The authors added these points in Lines 265-280, and they strengthen the manuscript.

However, the processes of carbonate dissolution and precipitation in the tomb walls need to be clarified. At the moment, there are a few contradicting statements about dissolution and precipitation in the bedrock. For example, in Lines 249-250 describing Macco facies, "In the inner walls of the intergranular voids microsparite cement precipitation and/or recrystallization often occurs, due to diagenetic processes of dissolution."

This sounds reasonable to me- the vadose zone often shows signs of dissolution followed by precipitation, as previously undersaturated waters become concentrated in Ca and CO3. However, in Lines 267-9 describing the general bedrock:

"Noteworthily, the characteristics of bedrock porosity indicate that the dissolution processes prevail on those of inorganic carbonate precipitation; indeed, meteoric cements as well as speleothems in the largest cavities are absent. Moreover, the bedrock where the tombs are carved is located in the shallow vadose zone (few meters below the surface) and it is barely susceptible to dissolution-reprecipitation processes,…"

*The reviewer is right, we deleted 'reprecipitation' because it can be misleading as reprecipitation can give the idea of development of concretions and abundant cements and this is not the case.*

In short, Section 3.1 says that the bedrock 1) contains evidence for dissolution and cement precipitation, 2) mostly evidence for dissolution with no cement precipitation, and 3) barely any dissolution or precipitation.

*Thank you for this comment, we deleted precipitation.*

These contradictions either need to be corrected, or if the authors are talking about different locations, that information needs to be clarified.
Here's why the discussion of dissolution and precipitation is important. A convincing argument for biogenic moonmilk requires: 1) a source of calcium, provided from calcite dissolution somewhere in the vadose zone, and 2) undersaturated meteoric waters which are less likely to precipitate calcite, requiring microbial metabolisms to foster precipitation. The authors address these points (end of 3.1), but there are two major questions left to address:
If the bedrock has no evidence for calcite dissolution, where is the calcium coming from to fuel biogenic moonmilk formation? On the other hand, if abiotic cements are present elsewhere in the bedrock, indicating periods of supersaturation, why does moonmilk need to be biogenic?
For publication to continue, the authors should:
1: clarify their interpretations on patterns of precipitation and dissolution in the bedrockis it an environment that contains both dissolution and precip? Is it dominated by dissolution? Or are both dissolution and precip limited?
*The porosity type (mainly vuggy and and moldic), the absence of mosaic cements (or other meteroric cements) points toward a dominance of dissolution processes instead of precipitation. On the other hand the main components are represented by coralline algae constituted by high magnesium calcite that is a metastable mineral phase of calcite. The original cabonate phase upon exposure to meteoric water will dissolve partially, increasing the availability of Ca2+ for microbial metabolism and mediated carbonate precipitation.*
2: once the authors have a consistent interpretation of precip/dissolution, clearly describe how such groundwaters create conditions where only microbes can make moonmilk (the

authors start to do this in Lines 278-279).
*Please, see the previous comment.*
3: (Optional) If the authors have any data on local groundwater, especially pH, Ca, and estimates for calcite saturation, that would greatly help build their case for a subsurface environment where only microbes could make moonmilk. This data could be from the authors themselves, or collected from the literature.

*We try to sample the groundwater in the rocks, but unfortunately the rock was dry because the summer weather conditions, but we will consider this comment for future experiments.*

2) Section 3.3- cultures
The strongest evidence for biogenic moonmilk at Tarquinia is presented in Line 329: "Indeed, under laboratory conditions, we have evidence that the grinded calcarenite, with its entire microbial community, when present in a medium containing urea and CaCl2 produced calcite. In the same conditions, calcium carbonate is not produced with sterile (autoclaved) grinded calcarenite (Benedetti et al. 2023)."
This is exactly the type of experiment needed to show the biogenicity of carbonate precipitation.
However, I have a few notes about the culture experiments and references.
First, Benedetti et al., 2023 is still in review. I don't know Biogeosciences' citation policies, but many journals require such references to be cited as (Benedetti et al. in review). More importantly, if this crucial data is rejected by the scientific community, then the argument for moonmilk biogenicity in this paper becomes much weaker.
This paper provides its' own culture experiments, as mentioned in Lines 304-329 and shown in Supp. Fig. 9. The methods for these experiments must be described in the Methods section- my apologies for missing that note last time. How much CaCl2 was used? How much urea? What was the temperature and carbonate saturation of the experiments? Most importantly, were sterilized experiments run during the research for this paper (not the unpublished work)? If so, sterilized comparisons would greatly help convince an audience that moonmilk is biogenic.
We added a new experiment, never published before, demonstrating that the microbial community is able to induce calcium carbonate precipitation.
The paper also needs to describe differences between the culturing experiments and the natural conditions in the bedrock. While it is impossible to exactly replicate the tomb environment inside a culture, the differences must still be addressed in the discussion. For example- do the authors think that Ca and urea are abundant in the bedrock environment? Especially important: What was the saturation of calcite in these experiments, and how does that relate to potential saturation states in the bedrock?
On a similar note, the authors provide a nice description of urea hydrolysis in the author reply. They mentioned leaving it out of the manuscript to avoid confusion, but I think this metabolism should be included in the discussion. Otherwise, the reader is missing crucial information on a potential process behind moonmilk formation.
*Thank you, we clarified this point in the text.*
For publication to continue, the authors should:
1: Describe culturing experiments in the Methods.
2: Change Benedetti references to "in review".
3: Provide extra discussion on the differences between culturing experiment conditions and conditions in the tombs- especially for chemical conditions (temperature, Ca, urea, calcite saturation).

4: Include a description of urea hydrolysis in the discussion. The best location is probably alongside the discussion of culturing experiments.

*Thank you, for the above suggestions, the text was updated.*

3) Section 3.3- RNA

The paper asserts that similar 16S RNA in bedrock and moonmilk are evidence for moonmilk biogenicity. While the results are interesting, and belong in the paper, such similarities alone are not enough to determine biogenicity, for one main reason.

There are other scenarios where moonmilk and bedrock communities are similar, but do not require biogenic moonmilk. For example, if the same microbes inhabited the entire porous bedrock sample before moonmilk formation, and then moonmilk formed abiotically through environmental change, both locations should have the same microbial populations. Therefore, similar populations can not distinguish biogenic vs abiogenic moonmilk.

When I brought this idea up in the previous round of reviews, the authors responded that the environment was unlikely to precipitate calcite abiotically. However, my issue here is not with environmental saturation- I've already addressed that topic. My issue is with the claim that similar RNA data can be used as evidence for biogenic moonmilk, as stated in Line 346:

These results show that in moonmilk and rocks the microbial composition is similar, irrespective of rock type (calcarenite and hybrid sandstone) or the environment where the samples were collected (outdoor or indoor). It should be noted that 16S SSU rRNA analysis does not provide information about metabolic activity, thus these data do not identify microorganisms that are active in CaCO3 deposition, but the overall data demonstrate that the endolytic community of the rocks is promoting moonmilk deposition.

In short, if the response to the question "Are similar communities in moonmilk and bedrock a biosignature?" is "Yes, but only because the environment cannot precipitate moonmilk", then the biosignature is not the RNA itself, but the presence of moonmilk in an undersaturated environment.

For publication to continue, the authors should:

1: Keep the RNA data and keep the comparisons of similar communities in the bedrock and moonmilk. However, any sentences that mention the RNA itself as evidence for biogenic moonmilk need to be removed (Lines 338-339, 350-352).

*We followed the suggestion of the referee, and we updated the text accordingly.*